# Phase I/II Study of LDE225 in Combination with Gemcitabine and Nab-Paclitaxel in Patients with Metastatic Pancreatic Cancer

**DOI:** 10.3390/cancers13194869

**Published:** 2021-09-28

**Authors:** Esther N. Pijnappel, Nienke P. M. Wassenaar, Oliver J. Gurney-Champion, Remy Klaassen, Koen van der Lee, Marjolein C. H. Pleunis-van Empel, Dick J. Richel, Marie C. Legdeur, Aart J. Nederveen, Hanneke W. M. van Laarhoven, Johanna W. Wilmink

**Affiliations:** 1Cancer Center Amsterdam, Department of Medical Oncology, Amsterdam University Medical Centers, University of Amsterdam, 1012 Amsterdam, The Netherlands; e.n.pijnappel@amsterdamumc.nl (E.N.P.); r.klaassen@amsterdamumc.nl (R.K.); k.s.vanderlee@amsterdamumc.nl (K.v.d.L.); d.j.richel@amsterdamumc.nl (D.J.R.); h.vanlaarhoven@amsterdamumc.nl (H.W.M.v.L.); 2Cancer Center Amsterdam, Department of Radiology, Amsterdam University Medical Centers, University of Amsterdam, 1012 Amsterdam, The Netherlands; n.p.wassenaar@amsterdamumc.nl (N.P.M.W.); o.j.gurney-champion@amsterdamumc.nl (O.J.G.-C.); a.j.nederveen@amsterdamumc.nl (A.J.N.); 3Department of Medical Oncology, Medisch Spectrum Twente, Twente, 7512 Enschede, The Netherlands; M.Pleunis@mst.nl (M.C.H.P.-v.E.); M.Legdeur@mst.nl (M.C.L.)

**Keywords:** pancreatic neoplasms, metastatic pancreatic ductal adenocarcinoma, Hedgehog signaling pathway inhibitor, LDE225, quantitative MRI

## Abstract

**Simple Summary:**

One of the reasons for treatment resistance of PDAC is the desmoplastic reaction initiating the production of large amounts of tumor stroma. LDE225 is a pharmacological Hedgehog signaling pathway inhibitor and is thought to specifically target tumor stroma. LDE225 in combination with gemcitabine and nab-paclitaxel was well-tolerated in patients with metastatic PDAC and has promising efficacy after prior treatment with FOLFIRINOX. Quantitative MRI suggested that LDE225 causes increased tumor diffusion and works particularly well in patients with poor baseline tumor perfusion. This suggests a clinical benefit of gemcitabine and nab-paclitaxel in combination with LDE225 in patients who received prior FOLFIRINOX. Future phase III clinical trials should confirm these results.

**Abstract:**

Background: Desmoplasia is a central feature of the tumor microenvironment in pancreatic ductal adenocarcinoma (PDAC). LDE225 is a pharmacological Hedgehog signaling pathway inhibitor and is thought to specifically target tumor stroma. We investigated the combined use of LDE225 and chemotherapy to treat PDAC patients. Methods: This was a multi-center, phase I/II study for patients with metastatic PDAC establishing the maximum tolerated dose of LDE225 co-administered with gemcitabine and nab-paclitaxel (phase I) and evaluating the efficacy and safety of the treatment combination after prior FOLFIRINOX treatment (phase II). Tumor microenvironment assessment was performed with quantitative MRI using intra-voxel incoherent motion diffusion weighted MRI (IVIM-DWI) and dynamic contrast-enhanced (DCE) MRI. Results: The MTD of LDE225 was 200 mg once daily co-administered with gemcitabine 1000 mg/m^2^ and nab-paclitaxel 125 mg/m^2^. In phase II, six therapy-related grade 4 adverse events (AE) and three grade 5 were observed. In 24 patients, the target lesion response was evaluable. Three patients had partial response (13%), 14 patients showed stable disease (58%), and 7 patients had progressive disease (29%). Median overall survival (OS) was 6 months (IQR 3.9–8.1). Blood plasma fraction (DCE) and diffusion coefficient (IVIM-DWI) significantly increased during treatment. Baseline perfusion fraction could predict OS (>222 days) with 80% sensitivity and 85% specificity. Conclusion: LDE225 in combination with gemcitabine and nab-paclitaxel was well-tolerated in patients with metastatic PDAC and has promising efficacy after prior treatment with FOLFIRINOX. Quantitative MRI suggested that LDE225 causes increased tumor diffusion and works particularly well in patients with poor baseline tumor perfusion.

## 1. Introduction

Pancreatic ductal adenocarcinoma (PDAC) is often a lethal condition and is ranked as the seventh highest cause of cancer related mortality in the world [1]. Of the newly diagnosed patients, 80–85% have locally advanced or metastatic disease [2]. Metastatic disease is characterized by a poor prognosis with a 5-year survival of less than 5% [2] and palliative chemotherapy is the only treatment option for this patient category [2].

The combination of 5-fluorouracil, leucovorin, irinotecan, and oxaliplatin (FOLFIRINOX) was the first major step forward in palliative systemic treatment since the introduction of gemcitabine monotherapy. Overall survival (OS) and quality of life (QOL) were significantly improved compared to gemcitabine monotherapy (11.5 vs. 6.8 months and QOL improvement of 10 points) [2,3,4,5]. Gemcitabine combined with nab-paclitaxel is one of the other currently used chemotherapy regimens. Survival was significantly improved with this combination compared to gemcitabine monotherapy (8.5 versus 6.7 months), while grade 3–4 toxicity was not increased [2,6,7]. Nowadays, FOLFIRINOX and gemcitabine with or without nab-paclitaxel are widely used first-line regimes, but there is limited evidence for second-line treatment for metastatic PDAC, especially after FOLFIRINOX [8,9].

One of the reasons for treatment resistance of PDAC is the desmoplastic reaction initiating the production of large amounts of tumor stroma [10,11,12,13]. Stroma limits the vascularization of tumor cells, which restricts the effective delivery of anti-cancer agents to the tumor [14]. The Hedgehog signaling pathway is known to be involved in tumor stroma formation in PDAC [14,15]. PDAC cells produce an increased amount of the Sonic Hedgehog ligand (SHh) [14,15,16,17,18]. By stimulating the patched 1 receptor, the ligand initiates the desmoplastic reaction, resulting in activation of the Hedgehog signaling pathway transcription factors Gli1,2,3 by Smoothened (SMO). Elevated production of the SHh ligand results in large amounts of tumor stroma and restricts vascularization [15,16,17,18]. A study in gemcitabine-resistant mouse models showed that by targeting the Hedgehog pathway, tumor vascularization increased, initiating higher efficacy of the chemotherapeutic treatment. Indeed, when combining gemcitabine with Hedgehog inhibition, tumor vasculature and subsequently gemcitabine delivery in the tumors were enhanced [13,19,20,21]. LDE225 is a pharmacological Hedgehog signaling pathway inhibitor and is thought to reduce the amount of tumor stroma.

Over the years, trials on Hedgehog inhibition (e.g., IPI-926, vismodegib) in combination with gemcitabine monotherapy, gemcitabine+nab-paclitaxel, or FOLFIRINOX showed no statistical difference in drug delivery or treatment efficacy [22,23]. However, there has not yet been a trial evaluating the effect of LDE225 in combination with gemcitabine+nab-paclitaxel.

In order to establish early signs of efficacy, we incorporated two tumor microenvironment imaging techniques in our study: Intravoxel-incoherent motion modelled diffusion-weighted magnetic resonance imaging (IVIM-DWI MRI) and dynamic contrast-enhanced (DCE) MRI. IVIM-DWI can non-invasively assess tumor diffusion and perfusion in vivo [24,25]. Low diffusion is typically associated with dense cell structures as in solid tumor and stroma whereas increased diffusion is associated with necrosis [26]. DCE MRI further probes the tumor’s micro vascularity and vascular permeability [27]. Our hypothesis is that a reduction in stroma caused by LDE225 leads to increased diffusion [28,29] and to revascularization of the tumor showing increased perfusion [30,31] and can be evaluated using these quantitative imaging techniques. Additionally, necrosis of the tumor as an overall result of the treatment is expected to show an increase in diffusion [28]. In a previous work, we already optimized IVIM-DWI and DCE MRI specifically for PDAC patients [32,33]. Using optimized pipelines, we correlated both IVIM-DWI and DCE MRI to pancreatic cancer pathology and treatment response in PDAC patients receiving surgery and illustrated the response in IVIM-DWI in patients receiving chemo-radiotherapy [26,34]. This highlights the potential of these techniques for evaluating treatment in PDAC patients.

Our current study is the first to explore the modification of the desmoplastic reaction seen in pancreatic cancer using two approaches, targeting tumor stroma by nab-paclitaxel and the Hedgehog inhibitor LDE225 and targeting the tumor cells with gemcitabine and nab-paclitaxel as a second-line treatment for patients with metastatic PDAC after first-line FOLFIRINOX.

## 2. Patients and Methods

### 2.1. Patient Population

Patients registered in this study were 18 years of age or older with histologically or cytologically confirmed diagnosis of metastatic PDAC and provided written informed consent. All patients had measurable disease on a pre-treatment CT scan according to response evaluation criteria in solid tumors (RECIST) 1.1, a World Health Organization (WHO, Geneva, Switzerland) performance status <2, and adequate bone marrow and organ function. Patients were excluded if they had a history of hypersensitivity to LDE225, or to drugs of similar chemical classes. Additionally, patients who underwent previous treatment with smoothened inhibitors or with known central nervous system (CNS) metastases were excluded.

### 2.2. Study Design and Treatment

This was a multi-center, open-label, interventional, noncontrolled, nonrandomized dose finding, phase I/II study, conducted in the Amsterdam University Medical Centers, location AMC in Amsterdam and in the Medical Spectrum Twente hospital in Enschede, both in the Netherlands. The study was approved by the ethical committee and registered at ClinicalTrials.gov with the identifying number NCT02358161. This study was conducted in agreement with the latest revision of the declaration of Helsinki and with the guidelines of good clinical practice issued by the European Union.

The objective of the phase I part of the study was to assess the safety, maximum tolerated dose (MTD), and dose limiting toxicities (DLTs) of LDE225 when co-administered with fixed doses of gemcitabine and nab-paclitaxel. The objective of the phase II part was to evaluate the efficacy and safety of the treatment combination after prior FOLFIRINOX treatment with response rates according to RECIST 1.1, median overall survival (OS) and progression free survival (PFS), changes in vascularity with DCE MRI, and changes in tumor stroma with DWI MRI.

At the start of our study, the largest study published on second-line treatment with gemcitabine and nab paclitaxel in patients with metastatic pancreatic cancer was the study by Portal et al. [35]. In this study, an objective response rate of 17% was seen. For the sample size calculation of the phase II part of the trial, we hypothesized that if the combination could lead to a response rate of 20%, developing a randomized trial is reasonable. With a power of 80% to detect such an increase and a significance level (alpha) of 0.10, the minimum sample size needed was 27 evaluable patients. Anticipating 10% of patients not being available for analysis, we planned to include a total number of 30 patients.

The starting dose of LDE225 was 400 mg daily dosed orally [36]. The doses for nab-paclitaxel and gemcitabine were 125 mg/m^2^ and 1000 mg/m^2^, respectively, administered weekly for three weeks every 4 weeks [6].

A DLT was defined as any dose limiting toxicity that was considered related to LDE225 alone or in combination with nab-paclitaxel and gemcitabine and unrelated to disease progression, inter-current illness, or concomitant medications (see Table 1). A minimum of three patients were entered on each dose level and followed for six weeks. Subsequent enrolment of new cohorts was based on the toxicity assessment in the first cycle and the documentation of any DLTs (see Table 2). If 0 out of 3 patients experienced a DLT at a given dose level, 3 patients were entered at the next dose level (+200 mg LDE225). When 1 out of 3 patients experienced a DLT, 3 patients were entered at the same dose level. Dose escalation was stopped when more than 2 patients experienced a DLT at a certain dose level. This dose level was declared the MTD.

### 2.3. Toxicity Assessment

Toxicity was graded using the common terminology criteria for adverse events (CTCAE) version 4.0. Before every treatment with gemcitabine and nab-paclitaxel, adverse events (AEs) were scored and reported in the case report file (CRF).

### 2.4. Tumor Response Evaluation

At baseline and subsequently every 8 weeks, tumor assessment and evaluation according to RECIST 1.1 was performed using CT-scan. (non)Target lesions were measured per organ side and documented in the CRF.

Overall response as well as response to (non)target lesions were described as complete response (CR), partial response (PR), stable disease (SD), or progressive disease (PD). If there were any new lesions compared to earlier screening, this was also documented in the case report file and regarded as PD.

IVIM-DWI and DCE MRI were performed at baseline (prior to treatment) and 8 weeks after the start of chemotherapy on a 3T MRI scanner (Ingenia, Philips, Best, The Netherlands). For IVIM-DWI MRI, a diffusion-weighted multi-slice echo-planar imaging sequence was used with TR/TE 2200/45 ms, respiratory triggering, and 12 b-values from 0 to 600 [33,34]. To minimize bowel movement, 20 mg of hyoscine bromide (Buscopan, Boehringer, Ingelheim, Germany) were administered intravenously before the acquisition. Detailed relevant MRI sequence parameters for all scans are given in Appendix A. The IVIM model was fitted to the signal decay of the DWI MRI as a function of the b-values using a bi-exponential fit to obtain the diffusion (D), pseudo diffusion (D*), and perfusion fraction (f) maps.

DCE MRI was performed identically to our previous work [32]. We acquired a dynamic series of 3-D spoiled gradient echo images with temporal resolution of 1.75 s, TR/TE 3.2/2.0 ms, and FA 20°. Scans were repeated for 280 s and after 10 dynamics, and 0.1 mmol/kg of 1.0 mmol/mL gadobutrol (Gadovist, Bayer Healthcare, Leverkusen, Germany) was injected intravenously at 5 mL/s followed by a 15 mL saline flush. Prior to the DCE acquisition, a Look-Locker ultrafast gradient echo was performed to assess the baseline T1 values, which were used to determine the contrast concentration. A population-based arterial input fraction was derived from another dataset of pancreatic cancer patients using the same scan settings and contrast administration protocol [32,37]. The Tofts model was fitted voxel-wise to acquire the extracellular extravascular space (EES) volume fraction (v_e_), the fractional plasma volume (v_p_), the transfer rate of contrast from plasma to EES (K^trans^), and the reflux rate of contrast from EES to plasma (k_ep_) [38].

The primary tumor was manually delineated on the baseline and post-treatment MRI scans using a 3D Slicer (Available online: http://www.slicer.org, accessed on 7 September 2021) under guidance of a contrast-enhanced MRI from the same scan session and a contrast-enhanced CT scan [39]. Cancerous pancreatic tissue was included into the region of interest (ROI) and biliary stents were excluded from the ROI. The mean parameter values of DCE and IVIM-DWI MRI from within the ROI were used for further analysis.

MRI data of patients from the phase I and phase II part of the trial were all combined to analyze the influence of LDE225 combined with gemcitabine and nab-paclitaxel on the characteristics of the tumor. A total of 36 patients underwent a baseline MRI scan of which 23 patients also underwent a post-treatment MRI scan (see Appendix A).

### 2.5. Statistical Analysis

Data in this study were analyzed using IBM SPSS software version 22. Baseline characteristics were described using mean (standard deviation) or median (interquartile range) for continuous variables and absolute number (percentage) for categorical variables. Evaluation of adverse events, safety, and efficacy of LDE225 combined with gemcitabine and nab-paclitaxel was performed with descriptive statistics. A Kaplan Meier analysis described the median OS between the different dose levels and treatment groups. Data analysis was anonymous. The probability of a type-I error was set at 0.05.

All statistical tests in the response evaluation using DCE and IVIM-DWI MRI were two-tailed and a significance level of α = 0.05 was used. The overall effect of the chemotherapy on the tumor was assessed by a Wilcoxon signed-rank test between MRI scans at baseline and post-treatment for all DCE and IVIM-DWI parameters. Subsequently, a receiver operating characteristics (ROC) analysis was performed to determine the specificity and sensitivity (using the Youden’s index) of baseline MRI parameters and the relative change in parameter value during treatment to predicting OS of PDAC patients receiving chemotherapy. The mean OS of 222 days was taken as a cut-off value to divide the patient group in long and short OS for the purpose of the ROC analysis.

The baseline and post-treatment CA 19.9 levels in combination with the relative change in MRI parameter values during treatment were also used to evaluate the treatment response. This was assessed with the spearman’s rank correlation coefficient.

## 3. Results

### 3.1. Phase I

#### 3.1.1. Characterization of the Study Cohort

In total, 39 patients were screened for eligibility between September 2014 and October 2016. In total, 13 patients were excluded, one patient because of gastrointestinal dysfunction, one patient because of the use of coumarin derivatives and CYP3A4/5 inhibitors, and the remaining 11 did not meet the inclusion criteria. A total of 26 patients were enrolled in the phase I part of this trial. For LDE225, there were eight dose reductions among six patients at various dose levels, all of them due to adverse events. Furthermore there were 12 temporary stops of LDE225 in six patients due to adverse events. Of the 26 patients, six patients had to discontinue the study due to adverse events, and 20 patients had to discontinue due to disease progression. Twenty-three patients were eligible for tumor response evaluation.

#### 3.1.2. MTD and DLT

Of the 26 patients that enrolled in the phase I part of study, one patient experienced a DLT at dose level 1. The DLT concerned diarrhea CTCAE grade 3 for more than 48 h, for which the patient had to discontinue study participation. This patient had received prior chemotherapy for metastatic disease. Moreover, the additional patients at this dose level that received prior chemotherapy for metastatic disease (i.e., FOLFIRINOX), the study treatment was less well tolerated with more grade 2/3 toxicities (11 grade 2/3 toxicities in the first month of treatment in three patients, compared with 7 grade 2/3 toxicities in the first month of treatment in three patients that had received no prior chemotherapy for metastatic disease). Therefore, we made the decision, with the approval of the local ethics committee, to split the study in two separate cohorts with individual dose-escalation schedules, based on whether or not patients had received prior chemotherapy for metastatic pancreatic cancer. We continued the dose-escalation schedule for the patients with no prior chemotherapy at dose level 1 and we de-escalated the dose level schedule for the previously treated patients to dose level-1.

The recommended phase II dose was 200 mg LDE225 in combination with gemcitabine 1000 mg/m^2^ and nab-paclitaxel 125 mg/m^2^ for patients treated with prior chemotherapy for metastatic disease (i.e., FOLFIRINOX) and 600 mg LDE225 in combination with gemcitabine 1000 mg/m^2^ and nab-paclitaxel 125 mg/m^2^ for patients that received no prior chemotherapy.

In phase II, we included the outcomes of the five patients of phase I that received 200 mg LDE225 and had prior treatment with FOLFIRINOX (*n* = 30 patients in total).

### 3.2. Phase II

#### 3.2.1. Characterization of the Study Cohort

We started the trial as a phase I study for patients with metastatic pancreatic cancer, in which patients could be included that were both chemotherapy naïve but also patients that had received prior FOLFIRINOX. Based on the results of the phase I part, we decided to continue in phase II with the patients that received prior FOLFIRINOX for metastatic disease. The reason for this decision was that in this patient cohort, despite the lower dose of LDE225, we saw responses, which was unprecedented at that time. Since FOLFIRINOX is the recommended first-line treatment for patients with metastatic PDAC in The Netherlands, there is a desperate need for a suitable second-line treatment after FOLFIRINOX failure. Therefore, we decided to continue with this patient cohort. Unfortunately, we do not have phase II data of the chemo-naïve group since we only included patients after first-line FOLFIRINOX failure in phase II.

In total, 33 patients were screened for eligibility between April 2017 and May 2018. Eight patients were excluded: seven did not meet the inclusion criteria and in one case, it was the decision of the patient. A total of 25 patients were enrolled in the phase II part of this trial. The baseline characteristics of the 25 patients, combined with the patients from phase I that received prior treatment with FOLFIRINOX and the LDE225 dosage of 200mg (*n* = 5), are depicted in Table 3. Patients were treated with a median number of two cycles (IQR 2–6). For LDE225, there was one dose reduction due to pneumonia and five temporary stops in four patients due to possible interaction with co-medication (1), hospitalization and LDE225 not present (1), and adverse events (3).

Patients discontinued treatment because of progressive disease (22), bacterial infection (1), sepsis (1), and diminished quality of life (1).

#### 3.2.2. Safety

Six therapy-related grade 4 adverse events (AEs) were observed: sepsis (2), neutropenia (2), elevated gamma GT (1), and thromboembolic event (1), and three therapy-related grade 5 AEs (sepsis (2) and pneumonia). Most common grade 3 therapy-related AEs were neutropenia (37%) and diarrhea (14.8%). The most frequently observed therapy-related AEs of any grade were fatigue 43(14%), thrombocytopenia 34(11%), diarrhea 28(9%), fever 26(8%), and vomiting 25(8%) (see Table 4).

#### 3.2.3. Tumor Response

In 24 patients, target lesion response was evaluable on CT scan. These 24 patients received a median of 3 (IQR 2.0–6.0) cycles and a median of 232 days (IQR 136.25–350.75) of study treatment. Tumor responses, defined as the percentage of change in target lesion volume of the best radiological response, are shown in Figure 1 as a waterfall plot. Three patients had partial response (13%), stable disease was seen in 14 patients (58%), and 7 patients had progressive disease (29%). Evaluation of progression and responses in days is shown in Figure 2 as a swimmers plot.

The median overall survival was 6.0 months (IQR 3.9–8.1). Median PFS was 4.0 months (IQR 1.2–6.7).

#### 3.2.4. MRI Analysis

All patients who received an MRI scan were included in the analysis (phase I and II part). This means that patients who were treated with other doses than 200 mg LDE225 were also included. In 36 patients, baseline MRI data were available for analysis of whom 23 patients also underwent a post-treatment MRI scan; however, we had to exclude one patient from the analysis because of major outliers in D (baseline D = 3.41 × 10^−3^ mm^2^/s) as a result of a necrotic tumor core at baseline. Therefore, we had 35 baseline and 22 post-treatment MRI scans in our analysis. However, one patient had no data on perfusion, diffusion, and pseudo diffusion and had to be excluded from these analyses too. The total number of patients with post-treatment IVIM-DWI MRI scans that were analyzable was 21. An example of the parameter maps can be seen in Figure 3. A significant increase of v_p_ and D was seen post-treatment compared to baseline values (see Table 5 and Figure 4).

There was no statistical difference in OS between different dose levels. The area under the curve (AUC), sensitivity, specificity, and cut off values are shown in Table 6. At baseline, the IVIM-DWI parameter f was most promising for predicting OS, with highest AUC of 0.85, with a sensitivity of 80% and a specificity of 85% (see Figure 5). Patients with low baseline perfusion (f < 5%) had the highest chance of having above-median OS. When assessing change in the parameter value over treatment, Δf gave the highest AUC value (0.786), with a sensitivity and specificity of 80% and 86%, respectively. Patients with an increase in perfusion during treatment (Δ > 16%) had the highest chance of above-median OS. Ten out of 21 patients had an increase in perfusion during treatment higher than 16% (Δ > 16%). The median OS of this subgroup was 291 days. CA 19.9 levels at baseline and at evaluation can be found in the Appendix A. We found a significant correlation between CA 19.9 levels and perfusion at evaluation r = −0.618 (*p* = 0.019) and a significant correlation between CA 19.9 and OS r = −0.487 (*p* = 0.026).

## 4. Discussion

In this phase I/II trial, which mainly focused on FOLFIRINOX-pretreated patients with metastatic pancreatic cancer, LDE225 in combination with gemcitabine and nab-paclitaxel demonstrated a manageable safety profile and promising efficacy. The overall response rate (ORR) and durability of response compares favorably with outcomes provided with currently available therapy for this population.

The reason for this focus was that in the patient cohort that was previously treated with FOLFIRINOX in the phase I part of the trial, despite the lower dose of LDE225, we saw responses, which was unprecedented at that time. Therefore, we decided to focus on the post-FOLRIRINOX group in the phase II part of the trial. Indeed, the evidence for second-line treatment after failure on FOLFIRINOX is scarce. Since FOLFIRINOX is the recommended first-line treatment for patients with metastatic PDAC in The Netherlands, there is a desperate need for a suitable second-line treatment after FOLFIRINOX.

There are a few randomized clinical trials in advanced pancreatic cancer, but they all have been conducted after first-line gemcitabine-based chemotherapy. The most promising combination in this setting is liposomal irinotecan in combination with 5-FU/LV, demonstrating a median survival of 6.1 months versus 4.2 months for the 5-FU/LV single agent [40]. After failure on FOLFIRINOX, data on second-line treatment are sparse. Although a gemcitabine-based regimen combined with nab-paclitaxel might be an option, randomized trials to confirm this suggestion are lacking. In the ACCORD/ PRODIGE 4 trial, about 50% of patients underwent second-line treatment with gemcitabine, with a median OS of 4.4 months, which is less favorable compared to an OS of 6 months in our clinical trial [4]. Other studies describing treatment with gemcitabine and nab-paclitaxel after FOLFIRINOX failure found lower median OS compared to our study [41,42,43]. Currently, there is no randomized evidence available on second-line treatment with gemcitabine and nab-paclitaxel after FOLFIRINOX failure. Observational cohort studies on second-line treatment with gemcitabine and nab-paclitaxel after FOLRIRINOX treatment in first line showed ORR of 13% and 17% [35,44,45]. These might be comparable to our ORR of 13%, but as opposed to other phase II/III studies on metastatic PDAC patients, median PFS and ORR in our study were higher [40,46,47,48]. The combination treatment of LDE225 with gemcitabine and nab-paclitaxel showed an improved biologic activity and was safely tolerated. However, the non-controlled design does not permit any conclusions, and future phase III clinical trials should confirm these results.

In The Netherlands, FOLFIRINOX is currently the recommended first-line treatment for patients with metastatic PDAC [8,49]. For patients who are not eligible for FOFIRINOX in first line, it would be interesting to preselect patients for LDE225 in combination with gemcitabine and nab-paclitaxel by using MRI (lower baseline perfusion fraction results in higher OS) in future studies.

The adverse events observed in our study were different from phase I studies with LDE225 monotherapy in patients with advanced solid tumors of any kind, including medulloblastoma and basal cell carcinoma. These studies most commonly found fatigue (2.3%), anorexia (2.3%), and elevated creatine phosphokinase (CPK) levels (4.7%) [36,50,51]. The difference in adverse events might be attributable to the addition of gemcitabine and/or nab-paclitaxel. However, although the incidence of adverse events is higher compared to previous studies, the toxicity was manageable enough for patients to continue treatment.

We were able to detect treatment effects from combined LDE225, gemcitabine, and nab-paclitaxel using quantitative MRI. We showed that the fractional plasma volume and diffusion of the tumor increased during treatment. Two mechanisms might contribute to this increase: apoptosis as a result of the chemotherapy reaching the tumor and the decrease of stroma due to LDE225. The lower cellularity due to these two processes causes a higher diffusion [28,52]. Various studies also described an increase of diffusion in tumors due to chemotherapy [53,54]. We excluded one patient from the analysis because of necrosis at baseline resulting in outliers in baseline D values. In this specific case, the response of the tumor to chemotherapy is expected to be different, and the necrotic cells will be cleared and less tumor cells will become necrotic.

Furthermore, we found that the baseline perfusion fraction can be used to predict OS. In patients with lower baseline perfusion fraction, the OS was higher. Additionally, an increase in perfusion fraction during treatment resulted in a better prognosis. These results can be explained by the treatment with LDE225, which specifically targets the tumor stroma. Patients with tumors that have a higher level of stroma at baseline will show a lower baseline perfusion fraction. The relative reduction of stroma by LDE225 will be higher in these patients than in patients with a lower amount of stroma at baseline. Our findings highlight the importance of assessing the tumor microenvironment with DCE and particularly IVIM-DWI during treatment. Furthermore, we showed that these techniques may allow for precision medicine by selecting patients most likely to benefit from LDE225.

A limitation of this study is that all patients who received MRI scans, also patients with LDE225 doses other than 200 mg, were included in the MRI analyses. Since all other analyses (e.g., on OS and PFS) were only performed on patients receiving 200 mg LDE225, there could be some discrepancy between these results. In addition, the studied patients is a very selected group, because metastatic PDAC with a WHO performance status of 0 or 1 after pre-treatment with FOLFIRINOX is remarkable.

## 5. Conclusions

In conclusion, this study showed that LDE225 in combination with gemcitabine and nab-paclitaxel as second line treatment is well-tolerated in patients with metastatic pancreatic cancer and has promising efficacy. The underlying mechanism of targeting stroma was validated in vivo. IVIM-DWI imaging may allow for selecting patients that could most benefit from LDE225 in the future.

## Figures and Tables

**Figure 1 cancers-13-04869-f001:**
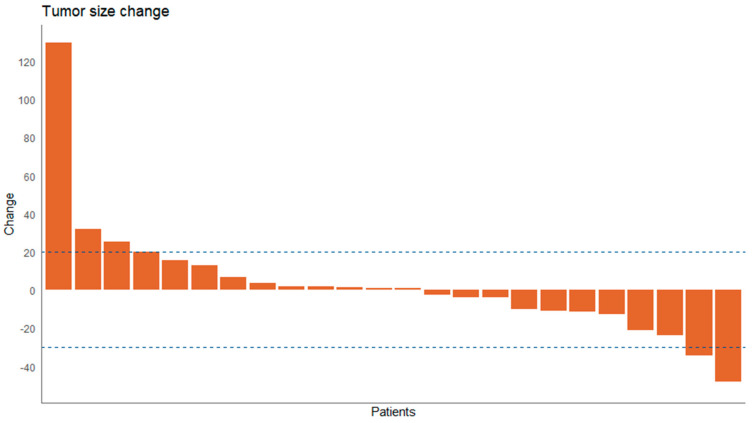
Waterfall plot of tumor response depicted as percentage of tumor volume change phase I and II combined for the patients of phase I that received prior treatment with FOLFIRINOX and treatment with LDE225 200 mg.

**Figure 2 cancers-13-04869-f002:**
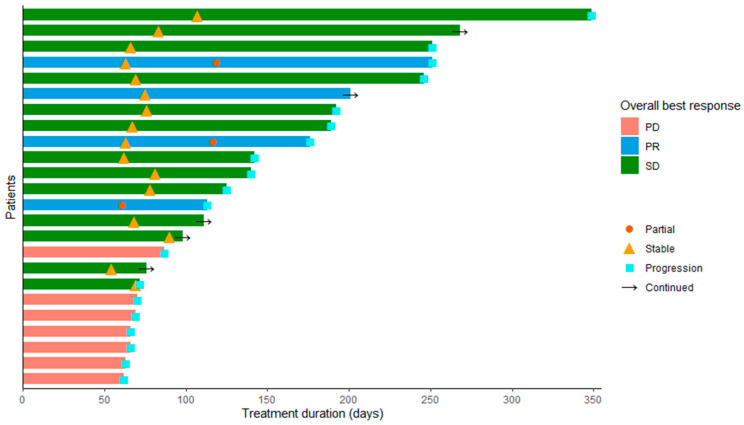
Swimmers plot for evaluation of progression and responses in days, responses as indicated phase I and II combined for the patients of phase I that received prior treatment with FOLFIRINOX and treatment with LDE225 200 mg. PD = progressive disease, PR = partial response, SD = stable disease.

**Figure 3 cancers-13-04869-f003:**
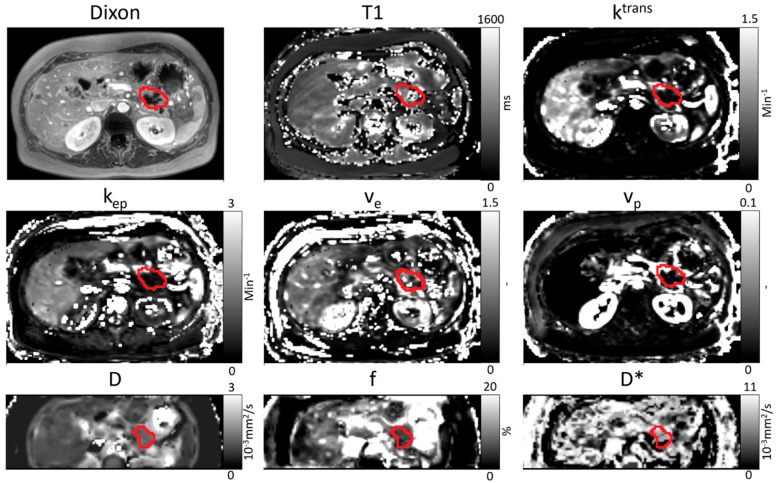
Example of an anatomical image (Dixon), T1 map, parameter maps of DCE MRI (ktrans, kep, ve, and vp), and parameter maps of IVIM-DWI MRI (D, f and D*) for one patient at baseline. The primary tumor is manually delineated in red. The delineation is performed separately for IVIM-DWI and DCE MRI scans.

**Figure 4 cancers-13-04869-f004:**
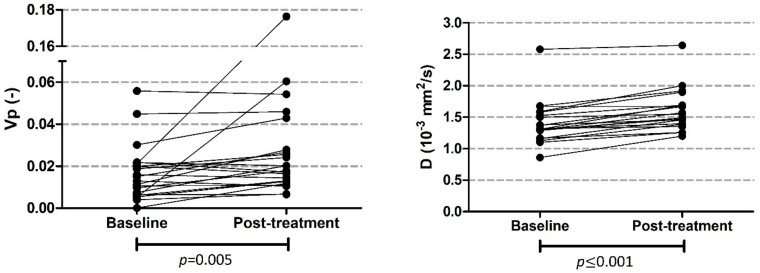
Plot of v_p_ and D at baseline and post-treatment. Both parameters significantly increased after treatment.

**Figure 5 cancers-13-04869-f005:**
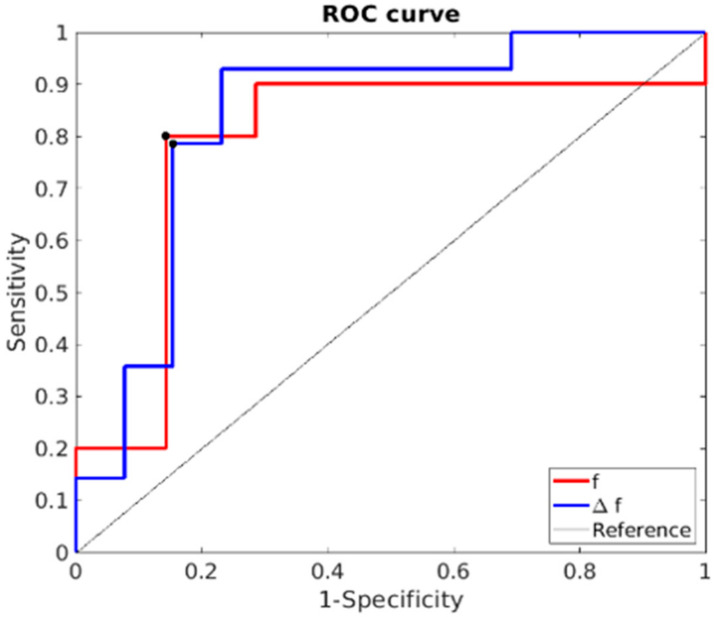
ROC curve of baseline f and Δf values for the prediction of OS > 222 days. The black dots indicate the cut-off values on which the sensitivity and specificity are determined. The AUC of f and Δf are 0.846 and 0.786, respectively.

**Table 1 cancers-13-04869-t001:** Criteria for defining dose-limiting toxicities (DLTs).

Toxicity	DLT Criteria
Toxicity leading to skipped/delayed dose	An AE (except for alopecia) of any grade, considered to be related to the study drug, leading to a dose interruption of more than 7 consecutive days, despite supportive treatment, will be considered to be a DLT.
Re-occurred toxicity	If the 2nd occurrence of an initially non-dose limiting toxicity (e.g., grade 1 neutropenia that resolved within 7 days at 1st occurrence) leads to a dose reduction within 42 days of the first dose of LDE225, this will be considered a DLT
Hematologic ^a^	CTCAE grade 4 neutropenia for >5 consecutive days
	CTCAE grade 4 thrombocytopenia CTCAE grade 3 with CTCAE grade > 2 bleeding
	CTCAE grade > 3 neutropenia with fever > 38.5 °C (non axillary)
Renal	≥CTCAE grade 3 serum creatinine
Hepatic	Total bilirubin ≥ 2.0× ULN to ≤ 3.0× ULN for >7 consecutive days. AST or ALT CTCAE grade ≥ 3 in conjunction with blood bilirubin CTCAE grade ≥ 2 of any duration. If not related to biliary obstruction/biliary stent dysfunction.
	≥CTCAE grade 3 total bilirubin. If not related to biliary obstruction/biliary stent dysfunction.
	CTCAE grade 3 AST or ALT for >7 consecutive days
	CTCAE grade 4 AST or ALT
Metabolic/Laboratory	CTCAE grade 3 asymptomatic amylase and/or lipase > 7 consecutive days
	CTCAE grade 4 asymptomatic amylase and/or lipase
Pancreatitis	≥CTCAE grade 2, if not related to biliary obstruction/stent dysfunction
Cardiac	Cardiac toxicity ≥ CTCAE grade 3 or cardiac event that is symptomatic or requires medical intervention
	QTcF > 500 ms confirmed by at least one ECG
	Clinical signs of cardiac disease, such as unstable angina or myocardial infarction, or Troponin ≥ CTCAE grade 3
Neurotoxicity	≥1 CTCAE grade level increase
Dematologic	≥CTCAE Grade 2 photosensitivity
	CTCAE Grade 3 rash for >7 consecutive days despite skin toxicity treatment
	CTCAE Grade 4 rash
Fatigue	≥CTCAE grade 3 for >7 consecutive days
	CTCAE grade 4
Other adverse events	≥CTCAE grade 3 adverse events (excluding ≥ CTCAE grade 3 lymphopenia or ≥ CTCAE grade 3 elevations in alkaline phosphatase
	≥CTCAE grade 3 vomiting/nausea ≥ 48 h, despite the use of anti-emetic therapy
	≥CTCAE grade 3 diarrhea ≥ 48 h, despite the use of anti-diarrheal therapy
CK elevation	≥CTCAE grade 3
Exception to DLT criteria	CTCAE grade 3 or 4 hypersensitivity or signs of allergic reaction

Whenever a DLT occurs: Study drug MUST be completely discontinued immediately. A single patient is assumed not to tolerate the dose if he/she experiences at least one DLT. ^a^ ≥CTCAE grade 3 anemia will not be considered DLT unless judged to be a hemolytic process secondary to study drug. ≥CTCAE grade 3 lymphopenia will not be considered DLT unless clinically significant.

**Table 2 cancers-13-04869-t002:** Dose escalation scheme phase 1.

Dose Level	LDE225	Gemcitabine	Nab-Paclitaxel	Minimum Number of Patients
−1	200 mg	1000 mg/m^2^	125 mg/m^2^	--
1 (starting)	400 mg	1000 mg/m^2^	125 mg/m^2^	3
2	600 mg	1000 mg/m^2^	125 mg/m^2^	3
3	800 mg	1000 mg/m^2^	125 mg/m^2^	3

**Table 3 cancers-13-04869-t003:** Patient characteristics phase I and II combined for the patients of phase I that received prior treatment with FOLFIRINOX and treatment LDE225 200 mg.

Variable	*n* = 30
Gender	
Male	17 (57%)
Female	13 (43%)
Age at start of study	62.1 (6.7)
WHO performance status at start study	
0	12 (40%)
1	16 (53%)
2	2 (7%)
Prior chemotherapy	30 (100%)
Prior surgery	11 (37%)
Median number of cycles	2 (2–6)
Median survival	6.0 months (3.9–8.1)

WHO; World Health Organization, Mean has standard deviation (SD) between brackets, median has interquartile range (IQR) between brackets, and number has percentages between brackets.

**Table 4 cancers-13-04869-t004:** Adverse events phase I and II combined (all the patients that received prior treatment with FOLFIRINOX and treatment with LDE225 200 mg); possible, probable or definitely treatment related.

Adverse Event	*n* (%)
Alopecia	17 (6)
Anemia	7 (2)
Anorexia	12 (4)
Bacterial infection	2 (1)
Chills	7 (3)
Constipation	1 (0.3)
Diarrhea	28 (8)
Dysgeusia	3 (1)
Edema limb	6 (2)
Epistaxis	2 (1)
Erythema multiform	1 (0.3)
Eye disorder other: decreased vision	1 (0.3)
Fatigue	43 (14)
Febrile neutropenia	2 (1)
Fever	26 (8)
Flu-like symptoms	4 (1)
Hematoma hands	1 (0.3)
Infection	1 (0.3)
Infusion related infection	6 (2)
Leukocytopenia	1 (0.3)
Malaise	3 (1)
Mucositis oral	11 (4)
Myalgia	3 (1)
Nail loss	1 (0.3)
Nausea	23 (7)
Neuropathy	11 (4)
Neutropenic fever	1 (0.3)
Neutropenia	15 (5)
Papulopustular rash	1 (0.3)
Rash	3 (1)
Rash acneiform	1 (0.3)
Rash, maculo popular	1 (0.3)
Sepsis	2 (1)
Stomatitis	1 (0.3)
Thrombocytopenia	35 (11)
Vomiting	25 (8)

**Table 5 cancers-13-04869-t005:** Median and IQR values for DCE and IVIM parameters at baseline and post-chemotherapy. The *p*-value of the Wilcoxon signed-rank test is also presented. v_p_ and D show a significant increase between baseline and post-treatment values.

		Median (IQR 25–75%)	Wilcoxon *p*-Value
K^trans^ (min^−1^)	Baseline	0.172 (0.113–0.295)	0.101
	Post	0.179 (0.113–0.301)	
k_ep_ (min^−1^)	Baseline	0.375 (0.287–0.469)	0.527
	Post	0.343 (0.261–0.427)	
v_e_ (-)	Baseline	0.581 (0.435–0.768)	0.961
	Post	0.623 (0.403–0.797)	
v_p_ (-)	Baseline	0.0159 (0.0075–0.0304)	0.005
	Post	0.0190 (0.0125–0.0317)	
T1 (ms)	Baseline	674 (554–877)	0.987
	Post	695 (590–871)	
D (10^−3^ mm^2^/s)	Baseline	1.35 (1.22–1.50)	<0.001
	Post	1.52 (1.39–1.69)	
f (%)	Baseline	5.1 (3.0–7.1)	0.279
	Post	5.1 (3.8–6.5)	
D* (10^−3^ mm^2^/s)	Baseline	24.0 (10.6–71.5)	0.165
	Post	45.4 (24.1–108.3)	

**Table 6 cancers-13-04869-t006:** Results of ROC analysis for all MRI parameters at baseline. The AUC, sensitivity, specificity, and cut off value determined with the Youden’s Index is given. If the baseline parameter meets the statement of the cut-off value, the OS is expected to be higher.

Parameter	Cut Off Value	Sensitivity	Specificity	AUC
Baseline				
K^trans^ (min^−1^)	≥0.181	62%	60%	0.600
k_ep_ (min^−1^)	≤0.345	67%	46%	0.508
v_e_ (−)	≥0.544	54%	60%	0.569
v_p_ (−)	≥0.0155	69%	60%	0.600
T1 (ms)	≤834	47%	85%	0.528
D (10^−3^ mm^2^/s)	≥1.32	69%	47%	0.574
f (%)	≤5.1	80%	85%	0.846
D* (10^−3^ mm^2^/s)	≤22.9	73%	77%	0.779
Parameter change				
ΔK^trans^ (%)	≤14	71%	70%	0.614
Δk_ep_ (%)	≤−8	71%	70%	0.714
Δv_e_ (%)	≥4	60%	71%	0.571
Δv_p_ (%)	≤60	86%	80%	0.743
ΔT1 (%)	≥−18	80%	43%	0.614
ΔD (%)	≤9	86%	50%	0.586
Δf (%)	≥16	80%	86%	0.786
ΔD* (%)	≥35	80%	57%	0.586

Δ indicates that we are looking at percent changes between baseline and post treatment.

## Data Availability

The data underlying this article will be shared on reasonable request to the corresponding author.

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
