# Peer review of "Phase I/II Study of LDE225 in Combination with Gemcitabine and Nab-Paclitaxel in Patients with Metastatic Pancreatic Cancer"

_cancers, 2021, doi:10.3390/cancers13194869_

Round 1

Reviewer 1 Report

The authors had answered my questions. 

Author Response

Thank you for your comments.

Reviewer 2 Report

I thank the authors for their revisions.

Some of the previous points were addressed but a few major questions were not addressed (in either the test or the response letter):

  1. Results:
    1. Phase II - it seems, based on the phase I, that the group with previous FOLFIRINOX treatment exhibited more toxicities and which required de-escalation to -1 (i.e 200 mg daily). Can you please explain why you chose to continue your assessment for this group specifically and not the chemotherapy naïve group (that had a better tolerability of 600 mg daily)? If Phase II data exists on the other group, please include it.
    2. Tumor response - Please include pre and post treatment CA19-9 levels and not only the correlation coefficient to other metrics. 
  2. Discussion:  
    1. Please address the decision the focus on the post-FOLFIRINOX group.

Round 2

Reviewer 2 Report

I thank the authors for their revised manuscript and for their clarifications.

This manuscript is a resubmission of an earlier submission. The following is a list of the peer review reports and author responses from that submission.

Round 1

Reviewer 1 Report

The study is to combine targeted therapy, LDE225 with abraxane and gemcitabine as the 2nd or later line treatment in pancreatic cancer. I have some questions for the study.

  1. Please explain the statistical consideration for thedetermination of case number in phase II part?
  2. The ORR for LDE225 plus abraxane and gemcitabine was 13%. Please find some other results under abraxane and gemcitabine as the 2nd line to compare the efficacies.
  3. In phase II, six therapy related grade 4 adverse events (AE) and three grade 5 were observed. were the three grade 5 AEs related to teatment?
  4. How many patients had an increase in perfusion during treatmet (Δ>16%). Please show the mOS of the subgroup.
  5. In figure 3, the anatomical image only showed in the pancreas. How was the change of parameters in other metastatic lesions, such as liver metastasis? In addition, the presented case with relative small primary tumor and please show another one with bigger tumor size to define tumor from stroma clearly.
  6. I suggest to apply the combination (LDE225+AG) to the first-line for further development by using MRI for pre-selection because abraxane and gemcitabine may be the most common first-line regimen for metastaic PC. What 's the author's opinion for the next step for this combination?

Reviewer 2 Report

I thank and congratulate the authors for this interesting study looking at the clinical impact of targeting the hedgehog pathway using the SMO inhibitor LDE225 in mPDAc patients receiving Gem-Abraxane.

Overall the paper is well written, with clear objective and rationale.

Specific comments:

  1. Abstract:
    1. Please state that the cohort sizes of each phase.
    2. Please consider mentioning phase II study design (interventional, noncontrolled, nonrandomized)
    3.  Please consider adding that LDE225 was dosed orally.
  2. Introduction:
    1. Please consider including some biological data on LDE225 (function, in-vivo data, use in other clinical studies)
  3. Results:
    1. Phase II - it seems, based on the phase I, that the group with previous FOLFIRINOX treatment exhibited more toxicities and which required de-escalation to -1 (i.e 200 mg daily). Can you please explain why you chose to continue your assessment for this group specifically and not the chemotherapy naïve group (that had a better tolerability of 600 mg daily)? If Phase II data exists on the other group, please include it.
    2. Tumor response - Were CA19-9 levels assessed as part of the routine follow-up? please include these, as these are commonly used to complete radiographic assessment of tumor response.
  4. Discussion:  
    1. Please address the decision the focus on the post-FOLFIRINOX group.
    2. The suggestion of a clinical benefit from the addition of LDE225 to the 2nd line Gen-Abraxane regimen is somewhat controversial -  the non-controlled design does not permit any actual conclusions regarding survival benefit. The authors cite several other studies with worse survival outcomes, however there are other studies with 2nd line Gem-Abraxane after FOLFIRINOX with better survival outcomes as well (and in this case the patients were also selected to have a good ECOG). It is safer to say that that the combination has shown an improved biologic activity, was safely tolerated and that the non-controlled design does not permit any conclusions.     
